# Selective inference for group-sparse linear models

**Fan Yang**
Department of Statistics
University of Chicago
fyang1@uchicago.edu

**Rina Foygel Barber**
Department of Statistics
University of Chicago
rina@uchicago.edu

**Prateek Jain**
Microsoft Research India
prajain@microsoft.com

**John Lafferty**
Depts. of Statistics and Computer Science
University of Chicago
lafferty@galton.uchicago.edu

## Abstract

We develop tools for selective inference in the setting of group sparsity, including the construction of confidence intervals and p-values for testing selected groups of variables. Our main technical result gives the precise distribution of the magnitude of the projection of the data onto a given subspace, and enables us to develop inference procedures for a broad class of group-sparse selection methods, including the group lasso, iterative hard thresholding, and forward stepwise regression. We give numerical results to illustrate these tools on simulated data and on health record data.

## 1 Introduction

Significant progress has been recently made on developing inference tools to complement the feature selection methods that have been intensively studied in the past decade [6, 5, 9]. The goal of selective inference is to make accurate uncertainty assessments for the parameters estimated using a feature selection algorithm, such as the lasso [12]. The fundamental challenge is that after the data have been used to select a set of coefficients to be studied, this selection event must then be accounted for when performing inference, using the same data. A specific goal of selective inference is to provide p-values and confidence intervals for the fitted coefficients. As the sparsity pattern is chosen using nonlinear estimators, the distribution of the estimated coefficients is typically non-Gaussian and multimodal, even under a standard Gaussian noise model, making classical techniques unusable for accurate inference. It is of particular interest to develop finite-sample, non-asymptotic results.

In this paper, we present new results for selective inference in the setting of group sparsity [15, 3, 10]. We consider the linear model $Y = \boldsymbol{X}\beta + \mathcal{N}(0, \sigma^2 \boldsymbol{I}_n)$ where $\boldsymbol{X} \in \mathbb{R}^{n \times p}$ is a fixed design matrix. In many applications, the $p$ columns or features of $\boldsymbol{X}$ are naturally grouped into blocks $\mathcal{C}_1, \ldots, \mathcal{C}_G \subseteq \{1, \ldots, p\}$. In the high dimensional setting, the working assumption is that only a few of the corresponding blocks of the coefficients $\beta$ contain nonzero elements; that is, $\beta_{\mathcal{C}_g} = 0$ for most groups $g$. This group-sparse model can be viewed as an extension of the standard sparse regression model. Algorithms for fitting this model, such as the group lasso [15], extend well-studied methods for sparse linear regression to this grouped setting.

In the group-sparse setting, recent results of Loftus and Taylor [9] give a selective inference method for computing p-values for each group chosen by a model selection method such as forward stepwise regression; selection via cross-validation was studied in [9]. More generally, the inference technique of [7] applies to any model selection method whose outcome can be described in terms of quadratic

conditions on $Y$. However, their technique cannot be used to construct confidence intervals for the selected coefficients or for the size of the effects of the selected groups.

Our main contribution in this work is to provide a tool for constructing confidence intervals as well as p-values for testing selected groups. In contrast to the (non-grouped) sparse regression setting, the confidence interval construction does not follow immediately from the p-value calculation, and requires a careful analysis of non-centered multivariate normal distributions. Our key technical result precisely characterizes the density of $\|\mathcal{P}_{\mathcal{L}}Y\|_2$ (the magnitude of the projection of $Y$ onto a given subspace $\mathcal{L}$), conditioned on a particular selection event. This "truncated projection lemma" is the group-wise analogue of the "polyhedral lemma" of Lee et al. [5] for the lasso. This technical result enables us to develop inference tools for a broad class of model selection methods, including the group lasso [15], iterative hard thresholding [1, 4], and forward stepwise group selection [14].

In the following section we frame the problem of group-sparse inference more precisely, and present previous results in this direction. We then state our main technical results; the proofs of the results are given in the supplementary material. In Section 3 we show how these results can be used to develop inferential tools for three different model selection algorithms for group sparsity. In Section 4 we give numerical results to illustrate these tools on simulated data, as well as on the California county health data used in previous work [9]. We conclude with a brief discussion of our work.

## 2 Main results: selective inference over subspaces

To establish some notation, we will write $\mathcal{P}_{\mathcal{L}}$ for the projection to any linear subspace $\mathcal{L} \subseteq \mathbb{R}^n$, and $\mathcal{P}_{\mathcal{L}}^{\perp}$ for the projection to its orthogonal complement. For $y \in \mathbb{R}^n$, $\mathrm{dir}_{\mathcal{L}}(y) = \frac{\mathcal{P}_{\mathcal{L}}y}{\|\mathcal{P}_{\mathcal{L}}y\|_2} \in \mathcal{L} \cap \mathbb{S}^{n-1}$ is the unit vector in the direction of $\mathcal{P}_{\mathcal{L}}y$. This direction is not defined if $\mathcal{P}_{\mathcal{L}}y = 0$.

We focus on the linear model $Y = \boldsymbol{X}\beta + \mathcal{N}(0, \sigma^2 \boldsymbol{I}_n)$, where $\boldsymbol{X} \in \mathbb{R}^{n \times p}$ is fixed and $\sigma^2 > 0$ is assumed to be known. More generally, our model is $Y \sim \mathcal{N}(\mu, \sigma^2 \boldsymbol{I}_n)$ with $\mu \in \mathbb{R}^n$ unknown and $\sigma^2$ known. For a given block of variables $\mathcal{C}_g \subseteq [p]$, we write $\boldsymbol{X}_g$ to denote the $n \times |\mathcal{C}_g|$ submatrix of $\boldsymbol{X}$ consisting of all features of this block. For a set $\mathcal{S} \subseteq [G]$ of blocks, $\boldsymbol{X}_{\mathcal{S}}$ consists of all features that lie in any of the blocks in $\mathcal{S}$.

When we refer to "selective inference," we are generally interested in the distribution of subsets of parameters that have been chosen by some model selection procedure. After choosing a set of groups $\mathcal{S} \subseteq [G]$, we would like to test whether the true mean $\mu$ is correlated with a group $\boldsymbol{X}_g$ for each $g \in \mathcal{S}$ after controlling for the remaining selected groups, i.e. after regressing out all the other groups, indexed by $\mathcal{S} \backslash g$. Thus, the following question is central to selective inference:

$$\text{Question}_{g,\mathcal{S}} : \text{ What is the magnitude of the projection of } \mu \text{ onto the span of } \mathcal{P}_{\boldsymbol{X}_{\mathcal{S} \backslash g}}^{\perp} \boldsymbol{X}_g? \qquad (1)$$

In particular, we are interested in a hypothesis test to determine if $\mu$ is orthogonal to this span, that is, whether block $g$ should be removed from the model with group-sparse support determined by $\mathcal{S}$; this is the question studied by Loftus and Taylor [9] for which they compute p-values. Alternatively, we may be interested in a confidence interval on $\|\mathcal{P}_{\mathcal{L}}\mu\|_2$, where $\mathcal{L} = \mathrm{span}(\mathcal{P}_{\boldsymbol{X}_{\mathcal{S} \backslash g}}^{\perp} \boldsymbol{X}_g)$. Since $\mathcal{S}$ and $g$ are themselves determined by the data $Y$, any inference on these questions must be performed "post-selection," by conditioning on the event that $\mathcal{S}$ is the selected set of groups.

### 2.1 Background: The polyhedral lemma

In the more standard sparse regression setting without grouped variables, after selecting a set $\mathcal{S} \subseteq [p]$ of features corresponding to columns of $\boldsymbol{X}$, we might be interested in testing whether the column $\boldsymbol{X}_j$ should be included in the model obtained by regressing $Y$ onto $\boldsymbol{X}_{\mathcal{S} \backslash j}$. We may want to test the null hypothesis that $\boldsymbol{X}_j^{\top} \mathcal{P}_{\boldsymbol{X}_{\mathcal{S} \backslash j}}^{\perp} \mu$ is zero, or to construct a confidence interval for this inner product.

In the setting where $\mathcal{S}$ is the output of the lasso, Lee et al. [5] and Tibshirani et al. [13] characterize the selection event as a polyhedron in $\mathbb{R}^n$: for any set $\mathcal{S} \subseteq [p]$ and any signs $s \in \{\pm 1\}^{\mathcal{S}}$, the event that the lasso (with a fixed regularization parameter $\lambda$) selects the given support with the given signs is equivalent to the event $Y \in \mathcal{A} = \{y : \boldsymbol{A}y < b\}$, where $\boldsymbol{A}$ is a fixed matrix and $b$ is a fixed vector, which are functions of $\boldsymbol{X}, \mathcal{S}, s, \lambda$. The inequalities are interpreted elementwise, yielding a convex polyhedron $\mathcal{A}$. To test the regression question described above, one then tests $\eta^{\top}\mu$ for a fixed unit vector $\eta \propto \mathcal{P}_{\boldsymbol{X}_{\mathcal{S} \backslash j}}^{\perp} \boldsymbol{X}_j$. The "polyhedral lemma", found in [5, Theorem 5.2] and [13, Lemma 2], proves that the distribution of $\eta^{\top}Y$, after conditioning on $\{Y \in \mathcal{A}\}$ and on $\mathcal{P}_{\eta}^{\perp}Y$, is given by a

truncated normal distribution, with density

$$f(r) \propto \exp\left\{-(r - \eta^\top \mu)^2 / 2\sigma^2\right\} \cdot \mathbf{1}\left\{a_1(Y) \le r \le a_2(Y)\right\}. \tag{2}$$

The interval endpoints $a_1(Y), a_2(Y)$ depend on $Y$ only through $\mathcal{P}_\eta^\perp Y$ and are defined to include exactly those values of $r$ that are feasible given the event $Y \in \mathcal{A}$. That is, the interval contains all values $r$ such that $r \cdot \eta + \mathcal{P}_\eta^\perp Y \in \mathcal{A}$.

Examining (2), we see that under the null hypothesis $\eta^\top \mu = 0$, this is a truncated *zero-mean* normal density, which can be used to construct a p-value testing $\eta^\top \mu = 0$. To construct a confidence interval for $\eta^\top \mu$, we can instead use (2) with nonzero $\eta^\top \mu$, which is a truncated *noncentral* normal density.

## 2.2 The group-sparse case

In the group-sparse regression setting, Loftus and Taylor [9] extend the work of Lee et al. [5] to questions where we would like to test $\mathcal{P}_\mathcal{L}\mu$, the projection of the mean $\mu$ to some potentially multi-dimensional subspace, rather than simply testing $\eta^\top \mu$, which can be interpreted as a projection to a one-dimensional subspace, $\mathcal{L} = \mathrm{span}(\eta)$. For a fixed set $\mathcal{A} \subseteq \mathbb{R}^n$ and a fixed subspace $\mathcal{L}$ of dimension $k$, Loftus and Taylor [9, Theorem 3.1] prove that, after conditioning on $\{Y \in \mathcal{A}\}$, on $\mathrm{dir}_\mathcal{L}(Y)$, and on $\mathcal{P}_\mathcal{L}^\perp Y$, under the null hypothesis $\mathcal{P}_\mathcal{L}\mu = 0$, the distribution of $\|\mathcal{P}_\mathcal{L}Y\|_2$ is given by a truncated $\chi_k$ distribution,

$$\|\mathcal{P}_\mathcal{L}Y\|_2 \sim (\sigma \cdot \chi_k \text{ truncated to } \mathcal{R}_Y) \text{ where } \mathcal{R}_Y = \left\{r : r \cdot \mathrm{dir}_\mathcal{L}(Y) + \mathcal{P}_\mathcal{L}^\perp Y \in \mathcal{A}\right\}. \tag{3}$$

In particular, this means that, if we would like to test the null hypothesis $\mathcal{P}_\mathcal{L}\mu = 0$, we can compute a p-value using the truncated $\chi_k$ distribution as our null distribution. To better understand this null hypothesis, suppose that we run a group-sparse model selection algorithm that chooses a set of blocks $\mathcal{S} \subseteq [G]$. We might then want to test whether some particular block $g \in \mathcal{S}$ should be retained in this model or removed. In that case, we would set $\mathcal{L} = \mathrm{span}(\mathcal{P}_{\mathbf{X}_{\mathcal{S}\setminus g}}^\perp \mathbf{X}_g)$ and test whether $\mathcal{P}_\mathcal{L}\mu = 0$.

Examining the parallels between this result and the work of Lee et al. [5], where (2) gives either a truncated zero-mean normal or truncated noncentral normal distribution depending on whether the null hypothesis $\eta^\top \mu = 0$ is true or false, we might expect that the result (3) of Loftus and Taylor [9] can extend in a straightforward way to the case where $\mathcal{P}_\mathcal{L}\mu \ne 0$. More specifically, we might expect that (3) might then be replaced by a truncated *noncentral* $\chi_k$ distribution, with its noncentrality parameter determined by $\|\mathcal{P}_\mathcal{L}\mu\|_2$. However, this turns out not to be the case. To understand why, observe that $\|\mathcal{P}_\mathcal{L}Y\|_2$ and $\mathrm{dir}_\mathcal{L}(Y)$ are the length and the direction of the vector $\mathcal{P}_\mathcal{L}Y$; in the inference procedure of Loftus and Taylor [9], they need to condition on the direction $\mathrm{dir}_\mathcal{L}(Y)$ in order to compute the truncation interval $\mathcal{R}_Y$, and then they perform inference on $\|\mathcal{P}_\mathcal{L}Y\|_2$, the length. These two quantities are independent for a centered multivariate normal, and therefore if $\mathcal{P}_\mathcal{L}\mu = 0$ then $\|\mathcal{P}_\mathcal{L}Y\|_2$ follows a $\chi_k$ distribution even if we have conditioned on $\mathrm{dir}_\mathcal{L}(Y)$. However, in the general case where $\mathcal{P}_\mathcal{L}\mu \ne 0$, we do not have independence between the length and the direction of $\mathcal{P}_\mathcal{L}Y$, and so while $\|\mathcal{P}_\mathcal{L}Y\|_2$ is marginally distributed as a noncentral $\chi_k$, this is no longer true after conditioning on $\mathrm{dir}_\mathcal{L}(Y)$.

In this work, we consider the problem of computing the distribution of $\|\mathcal{P}_\mathcal{L}Y\|_2$ after conditioning on $\mathrm{dir}_\mathcal{L}(Y)$, which is the setting that we require for inference. This leads to the main contribution of this work, where we are able to perform inference on $\mathcal{P}_\mathcal{L}\mu$ beyond simply testing the null hypothesis that $\mathcal{P}_\mathcal{L}\mu = 0$.

## 2.3 Key lemma: Truncated projections of Gaussians

Before presenting our key lemma, we introduce some further notation. Let $\mathcal{A} \subseteq \mathbb{R}^n$ be any fixed open set and let $\mathcal{L} \subseteq \mathbb{R}^n$ be a fixed subspace of dimension $k$. For any $y \in \mathcal{A}$, consider the set

$$\mathcal{R}_y = \{r > 0 : r \cdot \mathrm{dir}_\mathcal{L}(y) + \mathcal{P}_\mathcal{L}^\perp y \in \mathcal{A}\} \subseteq \mathbb{R}_+.$$

Note that $\mathcal{R}_y$ is an open subset of $\mathbb{R}_+$, and its construction does not depend on $\|\mathcal{P}_\mathcal{L}y\|_2$, but we see that $\|\mathcal{P}_\mathcal{L}y\|_2 \in \mathcal{R}_y$ by definition.

**Lemma 1** (Truncated projection). *Let $\mathcal{A} \subseteq \mathbb{R}^n$ be a fixed open set and let $\mathcal{L} \subseteq \mathbb{R}^n$ be a fixed subspace of dimension $k$. Suppose that $Y \sim \mathcal{N}(\mu, \sigma^2 \mathbf{I}_n)$. Then, conditioning on the values of $\mathrm{dir}_\mathcal{L}(Y)$ and $\mathcal{P}_\mathcal{L}^\perp Y$ and on the event $Y \in \mathcal{A}$, the conditional distribution of $\|\mathcal{P}_\mathcal{L}Y\|_2$ has density*[1]

$$f(r) \propto r^{k-1} \exp\left\{-\frac{1}{2\sigma^2}\left(r^2 - 2r \cdot \langle \mathrm{dir}_\mathcal{L}(Y), \mu \rangle\right)\right\} \cdot \mathbf{1}\{r \in \mathcal{R}_Y\}.$$

We pause to point out two special cases that are treated in the existing literature.

*Special case 1: $k = 1$ and $\mathcal{A}$ is a convex polytope.* Suppose $\mathcal{A}$ is the convex polytope $\{y : \boldsymbol{A}y < b\}$ for fixed $\boldsymbol{A} \in \mathbb{R}^{m \times n}$ and $b \in \mathbb{R}^m$. In this case, this almost exactly yields the "polyhedral lemma" of Lee et al. [5, Theorem 5.2]. Specifically, in their work they perform inference on $\eta^\top \mu$ for a fixed vector $\eta$; this corresponds to taking $\mathcal{L} = \text{span}(\eta)$ in our notation. Then since $k = 1$, Lemma 1 yields a truncated Gaussian distribution, coinciding with Lee et al. [5]'s result (2). The only difference relative to [5] is that our lemma implicitly conditions on $\text{sign}(\eta^\top Y)$, which is not required in [5].

*Special case 2: the mean $\mu$ is orthogonal to the subspace $\mathcal{L}$.* In this case, without conditioning on $\{Y \in \mathcal{A}\}$, we have $\mathcal{P}_\mathcal{L} Y = \mathcal{P}_\mathcal{L} \left( \mu + \mathcal{N}(0, \sigma^2 \boldsymbol{I}) \right) = \mathcal{P}_\mathcal{L} \left( \mathcal{N}(0, \sigma^2 \boldsymbol{I}) \right)$, and so $\|\mathcal{P}_\mathcal{L} Y\|_2 \sim \sigma \cdot \chi_k$. Without conditioning on $\{Y \in \mathcal{A}\}$ (or equivalently, taking $\mathcal{A} = \mathbb{R}^n$), the resulting density is then

$$ f(r) \propto r^{k-1} e^{-r^2/2\sigma^2} \cdot \mathbf{1}\left\{r > 0\right\} $$

which is the density of the $\chi_k$ distribution (rescaled by $\sigma$), as expected. If we also condition on $\{Y \in \mathcal{A}\}$ then this is a truncated $\chi_k$ distribution, as proved in Loftus and Taylor [9, Theorem 3.1].

### 2.4 Selective inference on truncated projections

We now show how the key result in Lemma 1 can be used for group-sparse inference. In particular, we show how to compute a p-value for the null hypothesis $H_0 : \mu \perp \mathcal{L}$, or equivalently, $H_0 : \|\mathcal{P}_\mathcal{L}\mu\|_2 = 0$. In addition, we show how to compute a one-sided confidence interval for $\|\mathcal{P}_\mathcal{L}\mu\|_2$, specifically, how to give a lower bound on the size of this projection.

**Theorem 1** (Selective inference for projections). *Under the setting and notation of Lemma 1, define*

$$ P = \frac{\int_{r \in \mathcal{R}_Y, r > \|\mathcal{P}_\mathcal{L} Y\|_2} r^{k-1} e^{-r^2/2\sigma^2} \, \mathsf{d}r}{\int_{r \in \mathcal{R}_Y} r^{k-1} e^{-r^2/2\sigma^2} \, \mathsf{d}r}. \tag{4} $$

*If $\mu \perp \mathcal{L}$ (or, more generally, if $\langle \text{dir}_\mathcal{L}(Y), \mu \rangle = 0$), then $P \sim \text{Uniform}[0, 1]$. Furthermore, for any desired error level $\alpha \in (0, 1)$, there is a unique value $L_\alpha \in \mathbb{R}$ satisfying*

$$ \frac{\int_{r \in \mathcal{R}_Y, r > \|\mathcal{P}_\mathcal{L} Y\|_2} r^{k-1} e^{-(r^2 - 2rL_\alpha)/2\sigma^2} \, \mathsf{d}r}{\int_{r \in \mathcal{R}_Y} r^{k-1} e^{-(r^2 - 2rL_\alpha)/2\sigma^2} \, \mathsf{d}r} = \alpha, \tag{5} $$

*and we have*

$$ \mathbb{P}\left\{\|\mathcal{P}_\mathcal{L}\mu\|_2 \geq L_\alpha\right\} \geq \mathbb{P}\left\{\langle \text{dir}_\mathcal{L}(Y), \mu \rangle \geq L_\alpha\right\} = 1 - \alpha. $$

*Finally, the p-value and the confidence interval agree in the sense that $P < \alpha$ if and only if $L_\alpha > 0$.*

From the form of Lemma 1, we see that we are actually performing inference on $\langle \text{dir}_\mathcal{L}(Y), \mu \rangle$. Since $\|\mathcal{P}_\mathcal{L}\mu\|_2 \geq \langle \text{dir}_\mathcal{L}(Y), \mu \rangle$, this means that any lower bound on $\langle \text{dir}_\mathcal{L}(Y), \mu \rangle$ also gives a lower bound on $\|\mathcal{P}_\mathcal{L}\mu\|_2$. For the p-value, the statement $\langle \text{dir}_\mathcal{L}(Y), \mu \rangle = 0$ is implied by the stronger null hypothesis $\mu \perp \mathcal{L}$. We can also use Lemma 1 to give a two-sided confidence interval for $\langle \text{dir}_\mathcal{L}(Y), \mu \rangle$; specifically, $\langle \text{dir}_\mathcal{L}(Y), \mu \rangle$ lies in the interval $[L_{\alpha/2}, L_{1-\alpha/2}]$ with probability $1 - \alpha$. However, in general this cannot be extended to a two-sided interval for $\|\mathcal{P}_\mathcal{L}\mu\|_2$.

To compare to the main results of Loftus and Taylor [9], their work produces the p-value (4) testing the null hypothesis $\mu \perp \mathcal{L}$, but does not extend to testing $\mathcal{P}_\mathcal{L}\mu$ beyond the null hypothesis, which the second part (5) of our main theorem is able to do.[2]

## 3 Applications to group sparse regression methods

In this section we develop inference tools for three methods for group-sparse model selection: forward stepwise regression (also considered by Loftus and Taylor [9] with results on hypothesis testing), iterative hard thresholding (IHT), and the group lasso.

## 3.1 General recipe

With a fixed design matrix, the outcome of any group-sparse selection method is a function of $Y$. For example, a forward stepwise procedure determines a particular sequence of groups of variables. We call such an outcome a *selection event*, and assume that the set of all selection events forms a countable partition of $\mathbb{R}^n$ into disjoint open sets: $\mathbb{R}^n = \cup_e \mathcal{A}_e$.[3] Each data vector $y \in \mathbb{R}^n$ determines a selection event, denoted $e(y)$, and thus $y \in \mathcal{A}_{e(y)}$.

Let $\mathcal{S}(y) \subseteq [G]$ be the set of groups selected for testing. This is assumed to be a function of $e(y)$, i.e. $\mathcal{S}(y) = \mathcal{S}_e$ for all $y \in \mathcal{A}_e$. For any $g \in \mathcal{S}_e$, let $\mathcal{L}_{e,g} = \mathrm{span}(\mathcal{P}^\perp_{\boldsymbol{X}_{\mathcal{S}_e \backslash g}} \boldsymbol{X}_g)$, the subspace of $\mathbb{R}^n$ indicating correlation with group $\boldsymbol{X}_g$ beyond what can be explained by the other selected groups.

Write $\mathcal{R}_Y = \{ r > 0 : r \cdot U + Y_\perp \in \mathcal{A}_{e(Y)} \}$, where $U = \mathrm{dir}_{\mathcal{L}_{e(Y),g}}(Y)$ and $Y_\perp = \mathcal{P}^\perp_{\mathcal{L}_{e(Y),g}} Y$. If we condition on the event $\{ Y \in \mathcal{A}_e \}$ for some $e$, then as soon as we have calculated the region $\mathcal{R}_Y \subseteq \mathbb{R}_+$, Theorem 1 will allow us to perform inference on the quantity of interest $\| \mathcal{P}_{\mathcal{L}_{e,g}} \mu \|_2$ by evaluating the expressions (4) and (5). In other words, we are testing whether $\mu$ is significantly correlated with the group $\boldsymbol{X}_g$, after controlling for all the other selected groups, $\mathcal{S}(Y) \backslash g = \mathcal{S}_e \backslash g$.

To evaluate these expressions accurately, ideally we would like an explicit characterization of the region $\mathcal{R}_Y \subseteq \mathbb{R}_+$. To gain a better intuition for this set, define $z_Y(r) = r \cdot U + Y_\perp \in \mathbb{R}^n$ for $r > 0$, and note that $z_Y(r) = Y$ when we plug in $r = \| \mathcal{P}_{\mathcal{L}_{e(Y),g}} Y \|_2$. Then we see that

$$\mathcal{R}_Y = \{ r > 0 : e(z_Y(r)) = e(Y) \}. \tag{6}$$

In other words, we need to find the range of values of $r$ such that, if we replace $Y$ with $z_Y(r)$, then this does not change the output of the model selection algorithm, i.e. $e(z_Y(r)) = e(Y)$. For the forward stepwise and IHT methods, we find that we can calculate $\mathcal{R}_Y$ explicitly. For the group lasso, we cannot calculate $\mathcal{R}_Y$ explicitly, but we can nonetheless compute the integrals required by Theorem 1 through numerical approximations. We now present the details for each of these methods.

## 3.2 Forward stepwise regression

Forward stepwise regression [2, 14] is a simple and widely used method. We will use the following version:[4] for design matrix $\boldsymbol{X}$ and response $Y = y$,

1. Initialize the residual $\widehat{\epsilon}_0 = y$ and the model $\mathcal{S}_0 = \varnothing$.
2. For $t = 1, 2, \ldots, T$,
   (a) Let $g_t = \arg\max_{g \in [G] \backslash \mathcal{S}_{t-1}} \{ \| \boldsymbol{X}_g^\top \widehat{\epsilon}_{t-1} \|_2 \}$.
   (b) Update the model, $\mathcal{S}_t = \{ g_1, \ldots, g_t \}$, and update the residual, $\widehat{\epsilon}_t = \mathcal{P}^\perp_{\boldsymbol{X}_{\mathcal{S}_t}} y$.

*Testing all groups at time $T$.* First we consider the inference procedure where, at time $T$, we would like to test each selected group $g_t$ for $t = 1, \ldots, T$; inference for this procedure was derived also in [8]. Our selection event $e(Y)$ is the ordered sequence $g_1, \ldots, g_T$ of selected groups. For a response vector $Y = y$, this selection event is equivalent to

$$\| \boldsymbol{X}_{g_k}^\top \mathcal{P}^\perp_{\boldsymbol{X}_{\mathcal{S}_{k-1}}} y \|_2 > \| \boldsymbol{X}_g^\top \mathcal{P}^\perp_{\boldsymbol{X}_{\mathcal{S}_{k-1}}} y \|_2 \text{ for all } k = 1, \ldots, T, \text{ for all } g \notin \mathcal{S}_k. \tag{7}$$

Now we would like to perform inference on the group $g = g_t$, while controlling for the other groups in $\mathcal{S}(Y) = \mathcal{S}_T$. Define $U, Y_\perp$, and $z_Y(r)$ as before. Then, to determine $\mathcal{R}_Y = \{ r > 0 : z_Y(r) \in \mathcal{A}_{e(Y)} \}$, we check whether all of the inequalities in (7) are satisfied with $y = z_Y(r)$: for each $k = 1, \ldots, T$ and each $g \notin \mathcal{S}_k$, the corresponding inequality of (7) can be expressed as

$$r^2 \cdot \| \boldsymbol{X}_{g_k}^\top \mathcal{P}^\perp_{\boldsymbol{X}_{\mathcal{S}_{k-1}}} U \|_2^2 + 2r \cdot \langle \boldsymbol{X}_{g_k}^\top \mathcal{P}^\perp_{\boldsymbol{X}_{\mathcal{S}_{k-1}}} U, \boldsymbol{X}_{g_k}^\top \mathcal{P}^\perp_{\boldsymbol{X}_{\mathcal{S}_{k-1}}} Y_\perp \rangle + \| \boldsymbol{X}_{g_k}^\top \mathcal{P}^\perp_{\boldsymbol{X}_{\mathcal{S}_{k-1}}} Y_\perp \|_2^2$$
$$> r^2 \cdot \| \boldsymbol{X}_g^\top \mathcal{P}^\perp_{\boldsymbol{X}_{\mathcal{S}_{k-1}}} U \|_2^2 + 2r \cdot \langle \boldsymbol{X}_g^\top \mathcal{P}^\perp_{\boldsymbol{X}_{\mathcal{S}_{k-1}}} U, \boldsymbol{X}_g^\top \mathcal{P}^\perp_{\boldsymbol{X}_{\mathcal{S}_{k-1}}} Y_\perp \rangle + \| \boldsymbol{X}_g^\top \mathcal{P}^\perp_{\boldsymbol{X}_{\mathcal{S}_{k-1}}} Y_\perp \|_2^2.$$

Solving this quadratic inequality over $r \in \mathbb{R}_+$, we obtain a region $\mathcal{I}_{k,g} \subseteq \mathbb{R}_+$ which is either a single interval or a union of two disjoint intervals, whose endpoints we can calculate explicitly with the quadratic formula. The set $\mathcal{R}_Y$ is then given by all values $r$ that satisfy the full set of inequalities:

$$\mathcal{R}_Y = \bigcap_{k=1,\ldots,T} \bigcap_{g \in [G] \backslash \mathcal{S}_k} \mathcal{I}_{k,g}.$$

This is a union of finitely many disjoint intervals, whose endpoints are calculated explicitly as above.

*Sequential testing.* Now suppose we carry out a sequential inference procedure, testing group $g_t$ at its time of selection, controlling only for the previously selected groups $\mathcal{S}_{t-1}$. In fact, this is a special case of the non-sequential procedure above, which shows how to test $g_T$ while controlling for $\mathcal{S}_T \backslash g_T = \mathcal{S}_{T-1}$. Applying this method at each stage of the algorithm yields a sequential testing procedure. (The method developed in [9] computes p-values for this problem, testing whether $\mu \perp \mathcal{P}^{\perp}_{\boldsymbol{X}_{\mathcal{S}_{t-1}}} \boldsymbol{X}_{g_t}$ at each time $t$.) See the supplementary material for detailed pseudo-code.

### 3.3 Iterative hard thresholding (IHT)

The iterative hard thresholding algorithm finds a $k$-group-sparse solution to the linear regression problem, iterating gradient descent steps with hard thresholding to update the model choice as needed [1, 4]. Given $k \geq 1$, number of iterations $T$, step sizes $\eta_t$, design matrix $\boldsymbol{X}$ and response $Y = y$,

1. Initialize the coefficient vector, $\beta_0 = 0 \in \mathbb{R}^p$ (or any other desired initial point).

2. For $t = 1, 2, \ldots, T$,

   (a) Take a gradient step, $\widetilde{\beta}_t = \beta_{t-1} - \eta_t \boldsymbol{X}^{\top}(\boldsymbol{X}\beta_{t-1} - y)$.

   (b) Compute $\|(\widetilde{\beta}_t)_{\mathcal{C}_g}\|_2$ for each $g \in [G]$ and let $\mathcal{S}_t \subseteq [G]$ index the $k$ largest norms.

   (c) Update the fitted coefficients $\beta_t$ via $(\beta_t)_j = (\widetilde{\beta}_t)_j \cdot \mathbf{1}\{j \in \cup_{g \in \mathcal{S}_t} \mathcal{C}_g\}$.

Here we are typically interested in testing Question$_{g,\mathcal{S}_T}$ for each $g \in \mathcal{S}_T$. We condition on the selection event, $e(Y)$, given by the sequence of $k$-group-sparse models $\mathcal{S}_1, \ldots, \mathcal{S}_T$ selected at each stage of the algorithm, which is characterized by the inequalities

$$\|(\widetilde{\beta}_t)_{\mathcal{C}_g}\|_2 > \|(\widetilde{\beta}_t)_{\mathcal{C}_h}\|_2 \quad \text{for all } t = 1, \ldots, T, \text{ and all } g \in \mathcal{S}_t, h \notin \mathcal{S}_t. \tag{8}$$

Fixing a group $g \in \mathcal{S}_T$ to test, determining $\mathcal{R}_Y = \{r > 0 : z_Y(r) \in \mathcal{A}_{e(Y)}\}$ involves checking whether all of the inequalities in (8) are satisfied with $y = z_Y(r)$. First, with the response $Y$ replaced by $y = z_Y(r)$, we show that we can write $\widetilde{\beta}_t = r \cdot c_t + d_t$ for each $t = 1, \ldots, T$, where $c_t, d_t \in \mathbb{R}^p$ are independent of $r$; in the supplementary material, we derive $c_t, d_t$ inductively as

$$\begin{cases} c_1 = \frac{\eta_1}{n}\boldsymbol{X}^{\top}U, \\ d_1 = (\boldsymbol{I} - \frac{\eta_1}{n}\boldsymbol{X}^{\top}\boldsymbol{X})\beta_0 + \frac{\eta_1}{n}\boldsymbol{X}^{\top}Y_{\perp}, \end{cases} \quad \begin{cases} c_t = (\boldsymbol{I}_p - \frac{\eta_t}{n}\boldsymbol{X}^{\top}\boldsymbol{X})\mathcal{P}_{\mathcal{S}_{t-1}}c_{t-1} + \frac{\eta_t}{n}\boldsymbol{X}^{\top}U, \\ d_t = (\boldsymbol{I}_p - \frac{\eta_t}{n}\boldsymbol{X}^{\top}\boldsymbol{X})\mathcal{P}_{\mathcal{S}_{t-1}}d_{t-1} + \frac{\eta_t}{n}\boldsymbol{X}^{\top}Y_{\perp}, \end{cases} \text{ for } t \geq 2.$$

Now we compute the region $\mathcal{R}_Y$. For each $t = 1, \ldots, T$ and each $g \in \mathcal{S}_t, h \notin \mathcal{S}_t$, the corresponding inequality in (8), after writing $\widetilde{\beta}_t = r \cdot c_t + d_t$, can be expressed as

$$r^2 \cdot \|(c_t)_{\mathcal{C}_g}\|_2^2 + 2r \cdot \langle (c_t)_{\mathcal{C}_g}, (d_t)_{\mathcal{C}_g} \rangle + \|(d_t)_{\mathcal{C}_g}\|_2^2 > r^2 \cdot \|(c_t)_{\mathcal{C}_h}\|_2^2 + 2r \cdot \langle (c_t)_{\mathcal{C}_h}, (d_t)_{\mathcal{C}_h} \rangle + \|(d_t)_{\mathcal{C}_h}\|_2^2.$$

As for the forward stepwise procedure, solving this quadratic inequality over $r \in \mathbb{R}_+$, we obtain a region $\mathcal{I}_{t,g,h} \subseteq \mathbb{R}_+$ that is either a single interval or a union of two disjoint intervals whose endpoints we can calculate explicitly. Finally, we obtain $\mathcal{R}_Y = \bigcap_{t=1,\ldots,T} \bigcap_{g \in \mathcal{S}_t} \bigcap_{h \in [G] \backslash \mathcal{S}_t} \mathcal{I}_{t,g,h}$.

### 3.4 The group lasso

The group lasso, first introduced by Yuan and Lin [15], is a convex optimization method for linear regression where the form of the penalty is designed to encourage group-wise sparsity of the solution. It is an extension of the lasso method [12] for linear regression. The method is given by

$$\widehat{\beta} = \arg\min_\beta \left\{ \tfrac{1}{2}\|y - \boldsymbol{X}\beta\|_2^2 + \lambda \sum_g \|\beta_{\mathcal{C}_g}\|_2 \right\},$$

where $\lambda > 0$ is a penalty parameter. The penalty $\sum_g \|\beta_{\mathcal{C}_g}\|_2$ promotes sparsity at the group level.[5]

For this method, we perform inference on the group support $\mathcal{S}$ of the fitted model $\widehat{\beta}$. We would like to test Question$_{g,\mathcal{S}}$ for each $g \in \mathcal{S}$. In this setting, for groups of size $\geq 2$, we believe that it is not possible to analytically calculate $\mathcal{R}_Y$, and furthermore, that there is no additional information that we can condition on to make this computation possible, without losing all power to do inference.

We thus propose a numerical approximation that circumvents the need for an explicit calculation of $\mathcal{R}_Y$. Examining the calculation of the p-value $P$ and the lower bound $L_\alpha$ in Theorem 1, we see that we can write $P = f_Y(0)$ and can find $L_\alpha$ as the unique solution to $f_Y(L_\alpha) = \alpha$, where

$$f_Y(t) = \frac{\mathbb{E}_{r \sim \sigma \cdot \chi_k}\left[e^{rt/\sigma^2} \cdot \mathbf{1}\{r \in \mathcal{R}_Y, r > \|\mathcal{P}_{\mathcal{L}}Y\|_2\}\right]}{\mathbb{E}_{r \sim \sigma \cdot \chi_k}\left[e^{rt/\sigma^2} \cdot \mathbf{1}\{r \in \mathcal{R}_Y\}\right]},$$

where we treat $Y$ as fixed in this calculation and set $k = \dim(\mathcal{L}) = \operatorname{rank}(\boldsymbol{X}_{\mathcal{S}\setminus g})$. Both the numerator and denominator can be approximated by taking a large number $B$ of samples $r \sim \sigma \cdot \chi_k$ and taking the empirical expectations. Checking $r \in \mathcal{R}_Y$ is equivalent to running the group lasso with the response replaced by $y = z_Y(r)$, and checking if the resulting selected model remains unchanged.

This may be problematic, however, if $\mathcal{R}_Y$ is in the tails of the $\sigma \cdot \chi_k$ distribution. We implement an importance sampling approach by repeatedly drawing $r \sim \psi$ for some density $\psi$; we find that $\psi = \|\mathcal{P}_{\mathcal{L}} Y\|_2 + \mathcal{N}(0, \sigma^2)$ works well in practice. Given samples $r_1, \ldots, r_B \sim \psi$ we then estimate

$$f_Y(t) \approx \widehat{f}_Y(t) := \frac{\sum_b \frac{\psi_{\sigma \cdot \chi_k}(r_b)}{\psi(r_b)} \cdot e^{r_b t / \sigma^2} \cdot \mathbf{1}\left\{r_b \in \mathcal{R}_Y, r_b > \|\mathcal{P}_{\mathcal{L}} Y\|_2\right\}}{\sum_b \frac{\psi_{\sigma \cdot \chi_k}(r_b)}{\psi(r_b)} \cdot e^{r_b t / \sigma^2} \cdot \mathbf{1}\left\{r_b \in \mathcal{R}_Y\right\}}$$

where $\psi_{\sigma \cdot \chi_k}$ is the density of the $\sigma \cdot \chi_k$ distribution. We then estimate $P \approx \widehat{P} = \widehat{f}_Y(0)$. Finally, since $\widehat{f}_Y(t)$ is continuous and strictly increasing in $t$, we estimate $L_\alpha$ by numerically solving $\widehat{f}_Y(t) = \alpha$.

## 4 Experiments

In this section we present results from experiments on simulated and real data, performed in R [11].[6]

### 4.1 Simulated data

We fix sample size $n = 500$ and $G = 50$ groups each of size 10. For each trial, we generate a design matrix $\boldsymbol{X}$ with i.i.d. $\mathcal{N}(0, 1/n)$ entries, set $\beta$ with its first 50 entries (corresponding to first $s = 5$ groups) equal to $\tau$ and all other entries equal to 0, and set $Y = \boldsymbol{X}\beta + \mathcal{N}(0, \boldsymbol{I}_n)$. We present the result for IHT here; the results for the other two methods can be found in the supplementary material.

We run IHT to select $k = 10$ groups over $T = 5$ iterations, with step sizes $\eta_t = 2$ and initial point $\beta_0 = 0$. For a moderate signal strength $\tau = 1.5$, we plot the p-values for each selected group in Figure 1; each group displays p-values only for those trials in which it was selected. The histogram of p-values for the $s$ true signals and for the $G - s$ nulls are also shown. We see that the the distribution of p-values for the true signals concentrates near zero while the null p-values are roughly uniform.

Next we look at the confidence intervals given by our method, examining their empirical coverage across different signal strengths $\tau$ in Figure 2. We fix confidence level 0.9 (i.e. $\alpha = 0.1$) and check empirical coverage with respect to both $\|\mathcal{P}_{\mathcal{L}}\mu\|_2$ and $\langle \operatorname{dir}_{\mathcal{L}}(Y), \mu \rangle$, with results shown separately for true signals and for nulls. For true signals, the confidence interval for $\|\mathcal{P}_{\mathcal{L}}\mu\|_2$ is somewhat conservative while the coverage for $\langle \operatorname{dir}_{\mathcal{L}}(Y), \mu \rangle$ is right at the target level, as expected from our theory. As signal strength $\tau$ increases, the gap is reduced for the true signals; this is because $\operatorname{dir}_{\mathcal{L}}(Y)$ becomes an increasingly more accurate estimate of $\operatorname{dir}_{\mathcal{L}}(\mu)$, and so the gap in the inequality $\|\mathcal{P}_{\mathcal{L}}\mu\|_2 \geq \langle \operatorname{dir}_{\mathcal{L}}(Y), \mu \rangle$ is reduced. For the nulls, if the set of selected groups contains the support of the true model, which is nearly always true for higher signal levels $\tau$, then the two are equivalent (as $\|\mathcal{P}_{\mathcal{L}}\mu\|_2 = \langle \operatorname{dir}_{\mathcal{L}}(Y), \mu \rangle = 0$), and coverage is at the target level. At low signal levels $\tau$, however, a true group is occasionally missed, in which case $\|\mathcal{P}_{\mathcal{L}}\mu\|_2 > \langle \operatorname{dir}_{\mathcal{L}}(Y), \mu \rangle$ strictly.

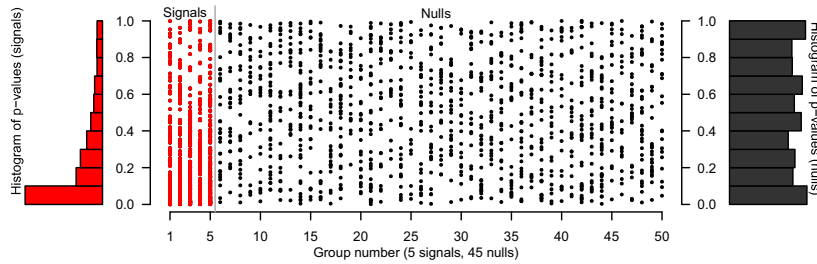

Figure 1: Iterative hard thresholding (IHT). For each group, we plot its p-value for each trial in which that group was selected, for 200 trials. Histograms of the p-values for true signals (left, red) and for nulls (right, gray) are attached.

### 4.2 California health data

We examine the 2015 California county health data[7] which was also studied by Loftus and Taylor [9]. We fit a linear model where the response is the log-years of potential life lost and the covariates

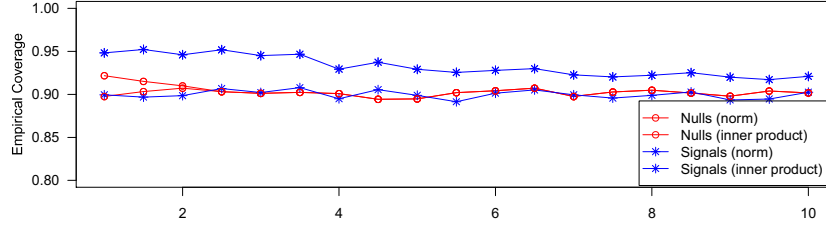

Figure 2: Iterative hard thresholding (IHT). Empirical coverage over 2000 trials with signal strength $\tau$. "Norm" and "inner product" refer to coverage of $\|\mathcal{P}_\mathcal{L}\mu\|_2$ and $\langle \mathrm{dir}_\mathcal{L}(Y), \mu\rangle$, respectively.

are the 34 predictors in this data set. We first let each predictor be its own group (i.e., group size 1) and run the three algorithms considered in Section 3. Next, we form a grouped model by expanding each predictor $\boldsymbol{X}_j$ into a group using the first three non-constant Legendre polynomials, $(\boldsymbol{X}_j, \frac{1}{2}(3\boldsymbol{X}_j^2-1), \frac{1}{2}(5\boldsymbol{X}_j^3-3\boldsymbol{X}_j))$. In each case we set parameters so that 8 groups are selected. The selected groups and their p-values are given in Table 1; interestingly, even when the same predictor is selected by multiple methods, its p-value can differ substantially across the different methods.

| Group size | Forward stepwise p-value / seq. p-value | | Iterative hard thresholding p-value | | Group lasso p-value | |
|---|---|---|---|---|---|---|
| 1 | 80th percentile income | 0.116 / 0.000 | 80th percentile income | 0.000 | 80th percentile income | 0.000 |
| | Injury death rate | 0.000 / 0.000 | Injury death rate | 0.000 | % Obese | 0.007 |
| | Violent crime rate | 0.016 / 0.000 | % Smokers | 0.004 | % Physically inactive | 0.040 |
| | % Receiving HbA1c | 0.591 / 0.839 | % Single-parent household | 0.009 | Violent crime rate | 0.055 |
| | % Obese | 0.481 / 0.464 | % Children in poverty | 0.332 | % Single-parent household | 0.075 |
| | Chlamydia rate | 0.944 / 0.975 | Physically unhealthy days | 0.716 | Injury death rate | 0.235 |
| | % Physically inactive | 0.654 / 0.812 | Food environment index | 0.807 | % Smokers | 0.701 |
| | % Alcohol-impaired | 0.104 / 0.104 | Mentally unhealthy days | 0.957 | Preventable hospital stays rate | 0.932 |
| 3 | 80th percentile income | 0.001 / 0.000 | Injury death rate | 0.000 | 80th percentile income | 0.000 |
| | Injury death rate | 0.044 / 0.000 | 80th percentile income | 0.000 | Injury death rate | 0.000 |
| | Violent crime rate | 0.793 / 0.617 | % Smokers | 0.000 | % Single-parent household | 0.038 |
| | % Physically inactive | 0.507 / 0.249 | % Single-parent household | 0.005 | % Physically inactive | 0.043 |
| | % Alcohol-impaired | 0.892 / 0.933 | Food environment index | 0.057 | % Obese | 0.339 |
| | % Severe housing problems | 0.119 / 0.496 | % Children in poverty | 0.388 | % Alcohol-impaired | 0.366 |
| | Chlamydia rate | 0.188 / 0.099 | Physically unhealthy days | 0.713 | % Smokers | 0.372 |
| | Preventable hospital stays rate | 0.421 / 0.421 | Mentally unhealthy days | 0.977 | Violent crime rate | 0.629 |

Table 1: Selective p-values for the California county health data experiment. The predictors obtained with forward stepwise are tested both simultaneously at the end of the procedure (first p-value shown), and also tested sequentially (second p-value shown), and are displayed in the selected order.

## 5   Conclusion

We develop selective inference tools for group-sparse linear regression methods, where for a data-dependent selected set of groups $\mathcal{S}$, we are able to both test each group $g \in \mathcal{S}$ for inclusion in the model defined by $\mathcal{S}$, and form a confidence interval for the effect size of group $g$ in the model. Our theoretical results can be easily applied to a range of commonly used group-sparse regression methods, thus providing an efficient tool for finite-sample inference that correctly accounts for data-dependent model selection in the group-sparse setting.

**Acknowledgments**

Research supported in part by ONR grant N00014-15-1-2379, and NSF grants DMS-1513594 and DMS-1547396.

## Footnotes

[1]Here and throughout the paper, we ignore the possibility that $Y \perp \mathcal{L}$ since this has probability zero.

[2]Their work furthermore considers the special case where the conditioning event, $Y \in \mathcal{A}$, is determined by a "quadratic selection rule," that is, $\mathcal{A}$ is defined by a set of quadratic constraints on $y \in \mathbb{R}^n$. However, extending to the general case is merely a question of computation (as we explore below for performing inference for the group lasso) and this extension should not be viewed as a primary contribution of this work.

[3]Since the distribution of $Y$ is continuous on $\mathbb{R}^n$, we ignore sets of measure zero without further comment.

[4]In practice, we would add some correction for the scale of the columns of $\boldsymbol{X}_g$ or for the number of features in group $g$; this can be accomplished with simple modifications of the forward stepwise procedure.

[5]Our method can also be applied to a modification of group lasso designed for overlapping groups [3] with a nearly identical procedure but we do not give details here.

[6]Code reproducing experiments: `http://www.stat.uchicago.edu/~rina/group_inf.html`

[7]Available at `http://www.countyhealthrankings.org`

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
