[Supplementary Material · select-group-suppmat.pdf]

# Supplementary material for:
# Selective inference for group-sparse linear models

**Fan Yang**
Department of Statistics
University of Chicago
Chicago IL 60637
fyang1@uchicago.edu

**Rina Foygel Barber**
Department of Statistics
University of Chicago
Chicago IL 60637
rina@uchicago.edu

**Prateek Jain**
Microsoft Research
Bangalore, India
prajain@microsoft.com

**John Lafferty**
Depts. of Statistics and Computer Science
University of Chicago
Chicago IL 60637
lafferty@galton.uchicago.edu

## A  Supplement

### A.1  Proof of Theorem 1

For any $y \in \mathcal{A}$, define a function $f_y : \mathbb{R} \to [0,1]$ as

$$f_y(t) = \frac{\int_{r \in \mathcal{R}_y, r > \|\mathcal{P}_{\mathcal{L}} y\|_2} r^{k-1} e^{-(r^2 - 2rt)/2\sigma^2} \; \mathrm{d}r}{\int_{r \in \mathcal{R}_y} r^{k-1} e^{-(r^2 - 2rt)/2\sigma^2} \; \mathrm{d}r}.$$

(As always we ignore the case $\mathcal{P}_{\mathcal{L}} y = 0$ to avoid degeneracy.)  By examining the integrals, we can immediately see that, for any fixed $y$, $f_y(t)$ is strictly increasing as a function of $t$, with $\lim_{t \to -\infty} f_y(t) = 0$ and $\lim_{t \to \infty} f_y(t) = 1$. These properties guarantee that, for any fixed $y$ and any fixed $\alpha \in (0,1)$, there is a unique $t \in \mathbb{R}$ with $f_y(t) = \alpha$, i.e. this proves the existence and uniqueness of $L_\alpha$ as required.

Furthermore, Lemma 1 immediately implies that, after conditioning on the event $Y \in \mathcal{A}$, and on the values of $\mathrm{dir}_{\mathcal{L}}(Y)$ and $\mathcal{P}_{\mathcal{L}}^{\perp} Y$, the conditional density of $\|\mathcal{P}_{\mathcal{L}} Y\|_2$ is

$$\propto r^{k-1} e^{-(r^2 - 2rt_Y)/2\sigma^2} \cdot \mathbf{1}\{r \in \mathcal{R}_Y\}$$

for $t_Y := \langle \mathrm{dir}_{\mathcal{L}}(Y), \mu \rangle$, and therefore, $f_Y(t_Y) \sim \mathrm{Uniform}[0,1]$. In the case that $\mu \perp \mathcal{L}$, we have $t_Y = 0$ always and therefore $P = f_Y(0) \sim \mathrm{Uniform}[0,1]$, as desired.  In the general case, by definition of $L_\alpha$, we have $f_Y(L_\alpha) = \alpha$ and so, again using the fact that $f_Y(\cdot)$ is strictly increasing,

$$f_Y(t_Y) \leq \alpha = f_Y(L_\alpha) \;\Leftrightarrow\; t_Y \leq L_\alpha,$$

and so by definition of $t_Y$,

$$\mathbb{P}\left\{ \langle \mathrm{dir}_{\mathcal{L}}(Y), \mu \rangle < L_\alpha \right\} = \mathbb{P}\left\{ t_Y < L_\alpha \right\} = \mathbb{P}\left\{ f_Y(t_Y) < \alpha \right\} = \alpha.$$

Furthermore, we know that $\|\mathcal{P}_{\mathcal{L}} \mu\|_2 \geq \langle \mathrm{dir}_{\mathcal{L}}(Y), \mu \rangle = t_Y$, and so

$$\mathbb{P}\left\{ \|\mathcal{P}_{\mathcal{L}} \mu\|_2 < L_\alpha \right\} \leq \mathbb{P}\left\{ \langle \mathrm{dir}_{\mathcal{L}}(Y), \mu \rangle < L_\alpha \right\} = \alpha.$$

Finally, we see that since $P = f_Y(0)$ while $\alpha = f_Y(L_\alpha)$, $P < \alpha$ if and only if $0 < L_\alpha$.

## A.2 Proof of Lemma 1

We begin with the following elementary calculation (for completeness the proof is given below):

**Lemma A.1.** *Suppose that* $\widetilde{Y} \sim \mathcal{N}(\widetilde{\mu}, \sigma^2 \boldsymbol{I}_k)$. *Let* $R = \|\widetilde{Y}\|_2 \in \mathbb{R}_+$ *and* $U = \operatorname{dir}(\widetilde{Y}) \in \mathbb{S}^{k-1}$ *be the radius and direction of the random vector* $\widetilde{Y}$. *Then the joint distribution of* $(R, U)$ *has density*

$$f(r, u) \propto r^{k-1} \exp\left\{-\frac{1}{2\sigma^2}\left(r^2 - 2r \cdot \langle u, \widetilde{\mu}\rangle\right)\right\} \text{ for } (r, u) \in \mathbb{R}_+ \times \mathbb{S}^{k-1}.$$

Next, let $\mathbf{V} \in \mathbb{R}^{n \times k}$ be an orthonormal basis for $\mathcal{L}$ and let

$$\widetilde{Y} = \mathbf{V}^\top Y \sim \mathcal{N}(\widetilde{\mu}, \sigma^2 \boldsymbol{I}_k) \text{ where } \widetilde{\mu} = \mathbf{V}^\top \mu.$$

Now let $R = \|\widetilde{Y}\|_2 = \|\mathcal{P}_{\mathcal{L}} Y\|_2$ and let $U = \operatorname{dir}(\widetilde{Y}) = \mathbf{V}^\top \operatorname{dir}_{\mathcal{L}}(Y)$; note that $\operatorname{dir}_{\mathcal{L}}(Y) = \mathbf{V}U$.

Defining $W = \mathcal{P}_{\mathcal{L}}^\perp Y$, we see that $Y = r \cdot \mathbf{V}u + w$, and that $\widetilde{Y} \perp\!\!\!\perp W$ by properties of the normal distribution. Combining this with the result of Lemma A.1, we see that the joint density of $(R, U, W)$ is given by

$$f_{R,U,W}(r, u, w) \propto r^{k-1} \exp\left\{-\frac{1}{2\sigma^2}\left(r^2 - 2r \cdot \langle u, \widetilde{\mu}\rangle\right)\right\} \cdot \exp\left\{-\frac{1}{2\sigma^2}\|w - \mathcal{P}_{\mathcal{L}}^\perp \mu\|_2^2\right\}$$

for $(r, u, w) \in \mathbb{R}_+ \times \mathbb{S}^{k-1} \times \mathcal{L}_\perp$. After conditioning on the event $\{Y \in \mathcal{A}\}$, this density becomes

$$\propto r^{k-1} \exp\left\{-\frac{1}{2\sigma^2}\left(r^2 - 2r \cdot \langle u, \widetilde{\mu}\rangle\right)\right\} \cdot \exp\left\{-\frac{1}{2\sigma^2}\|w - \mathcal{P}_{\mathcal{L}}^\perp \mu\|_2^2\right\} \cdot \mathbf{1}\{r \cdot \mathbf{V}u + w \in \mathcal{A}\}.$$

Next note that the event $\{Y \in \mathcal{A}\}$ is equivalent to $\{R \in \mathcal{R}_Y\}$ where $\mathcal{R}_Y = \{r > 0 : r \cdot \mathbf{V}U + W \in \mathcal{A}\}$, and so the conditional density of $R$, after conditioning on $U$, $W$, and on the event $Y \in \mathcal{A}$, is

$$\propto r^{k-1} \exp\left\{-\frac{1}{2\sigma^2}\left(r^2 - 2r \cdot \langle U, \widetilde{\mu}\rangle\right)\right\} \cdot \mathbf{1}\{R \in \mathcal{R}_Y\},$$

as desired. Now we prove our supporting result, Lemma A.1.

*Proof of Lemma A.1.* It's easier to work with the parametrization $(Z, U)$ where $Z = \log(R)$. By a simple change of variables calculation, the claim in the lemma is equivalent to showing that

$$f_{Z,U}(z, u) \propto e^{kz} \exp\left\{-\frac{1}{2\sigma^2}\left(e^{2z} - 2e^z \cdot \langle u, \widetilde{\mu}\rangle\right)\right\} \text{ for } (z, u) \in \mathbb{R} \times \mathbb{S}^{k-1}.$$

Fix any $\varepsilon \in (0, 1)$ and, for each $(z, u) \in \mathbb{R} \times \mathbb{S}^{k-1}$, consider the region $(z - \varepsilon, z + \varepsilon) \times \mathcal{C}_u^\varepsilon \subseteq \mathbb{R} \times \mathbb{S}^{k-1}$, where $\mathcal{C}_u^\varepsilon$ is a spherical cap, $\mathcal{C}_u^\varepsilon := \{v \in \mathbb{S}^{k-1} : \|v - u\|_2 < \varepsilon\}$. Let $s_\varepsilon$ be the surface area of $\mathcal{C}_u^\varepsilon \subseteq \mathbb{S}^{k-1}$ (note that this surface area does not depend on $u$ since it's rotation invariant).

To check that our density is correct, it is sufficient to check that

$$\mathbb{P}\left\{(Z, U) \in (z - \varepsilon, z + \varepsilon) \times \mathcal{C}_u^\varepsilon\right\} \propto$$
$$\text{Volume}\left((z - \varepsilon, z + \varepsilon) \times \mathcal{C}_u^\varepsilon\right) \cdot e^{kz} \cdot \exp\left\{-\frac{1}{2\sigma^2}\left(e^{2z} - 2e^z \cdot \langle u, \widetilde{\mu}\rangle\right)\right\} \cdot (1 + o(1)),$$

where the $o(1)$ term is with respect to the limit $\varepsilon \to 0$ while $(z, u)$ is held fixed, and where the constant of proportionality is independent of $\varepsilon$ and of $z, u$. We can also calculate $\text{Volume}\left((z - \varepsilon, z + \varepsilon) \times \mathcal{C}_u^\varepsilon\right) = 2\varepsilon \cdot s_\varepsilon$.

Now consider

$$\mathcal{Y}_{z,u}^\varepsilon = \left\{y \in \mathbb{R}^n : \frac{y}{\|y\|_2} \in \mathcal{C}_u^\varepsilon, \log(\|y\|_2) \in (z - \varepsilon, z + \varepsilon)\right\} \subseteq \mathbb{R}^n.$$

We have

$$\mathbb{P}\left\{(Z, U) \in (z - \varepsilon, z + \varepsilon) \times \mathcal{C}_u^\varepsilon\right\} = \mathbb{P}\left\{\widetilde{Y} \in \mathcal{Y}_{z,u}^\varepsilon\right\}.$$

Since $\mathcal{Y}_{z,u}^{\varepsilon} = \cup_{t\in(e^{z-\varepsilon},e^{z+\varepsilon})}(t\cdot\mathcal{C}_u^{\varepsilon})$, and the surface area of $t\cdot C_u^{\varepsilon} \subseteq t\cdot\mathbb{S}^{k-1}$ is equal to $s_\varepsilon t^{k-1}$, we can also calculate

$$\text{Volume}(\mathcal{Y}_{z,u}^{\varepsilon}) = \int_{t=e^{z-\varepsilon}}^{e^{z+\varepsilon}} s_\varepsilon t^{k-1}\,\mathsf{d}t = \frac{1}{k}s_\varepsilon t^k\Big|_{t=e^{z-\varepsilon}}^{e^{z+\varepsilon}} = \frac{1}{k}s_\varepsilon\cdot(e^{k(z+\varepsilon)}-e^{k(z-\varepsilon)}) = 2\varepsilon\cdot s_\varepsilon\cdot e^{kz}\cdot(1+o(1)),$$

since $e^{k(z+\varepsilon)} - e^{k(z-\varepsilon)} = e^{kz}\cdot 2k\varepsilon\cdot(1+o(1))$. And, since $\max_{y\in\mathcal{Y}_{z,u}^{\varepsilon}}\|y-e^z\cdot u\|_2 \to 0$ as $\varepsilon\to 0$, then for any $y\in\mathcal{Y}_{z,u}^{\varepsilon}$, the density of $\widetilde{Y}$ at this point is given by

$$f_{\widetilde{Y}}(y) = \frac{1}{\sqrt{(2\pi\sigma^2)^n}}e^{-\frac{1}{2\sigma^2}\|y-\widetilde{\mu}\|_2^2} = \frac{1}{\sqrt{(2\pi\sigma^2)^n}}e^{-\frac{1}{2\sigma^2}\|e^z\cdot u-\widetilde{\mu}\|_2^2}\cdot(1+o(1)),$$

where again the $o(1)$ term is with respect to the limit $\varepsilon\to 0$ while $(z,u)$ is held fixed. So, we have

$$\mathbb{P}\left\{(Z,U)\in(z-\varepsilon,z+\varepsilon)\times\mathcal{C}_u^{\varepsilon}\right\} = \mathbb{P}\left\{\widetilde{Y}\in\mathcal{Y}_{z,u}^{\varepsilon}\right\} = \int_{y\in\mathcal{Y}_{z,u}^{\varepsilon}} f_{\widetilde{Y}}(y)\,\mathsf{d}y$$

$$= \text{Volume}(\mathcal{Y}_{z,u}^{\varepsilon})\cdot\frac{1}{\sqrt{(2\pi\sigma^2)^n}}e^{-\frac{1}{2\sigma^2}\|e^z\cdot u-\widetilde{\mu}\|_2^2}\cdot(1+o(1))$$

$$= 2\varepsilon\cdot s_\varepsilon\cdot e^{kz}\cdot(1+o(1))\cdot\frac{1}{\sqrt{(2\pi\sigma^2)^n}}e^{-\frac{1}{2\sigma^2}\|e^z\cdot u-\widetilde{\mu}\|_2^2}\cdot(1+o(1))$$

$$= 2\varepsilon\cdot s_\varepsilon\cdot e^{kz}\cdot\exp\left\{-\frac{1}{2\sigma^2}\left(e^{2z}-2e^z\cdot\langle u,\widetilde{\mu}\rangle\right)\right\}\cdot\left[\frac{1}{\sqrt{(2\pi\sigma^2)^n}}e^{-\frac{1}{2\sigma^2}\|\widetilde{\mu}\|_2^2}\right]\cdot(1+o(1)),$$

which gives the desired result since the term in square brackets is constant with respect to $z,u,\varepsilon$. $\qquad\square$

### A.3 Derivations for IHT inference

Here we derive the formulas for the coefficients $c_t,d_t$ used in the inference procedure for group-sparse IHT. First, at time $t=1$,

$$\widetilde{\beta}_1 = b_0 - \eta_1\nabla f(b_0) = b_0 - \eta_1\left(\frac{1}{n}\boldsymbol{X}^\top(\boldsymbol{X}b_0 - z_Y(r))\right)$$

$$= r\cdot\left[\frac{\eta_1}{n}\boldsymbol{X}^\top U\right] + \left[(\boldsymbol{I}-\frac{\eta_1}{n}\boldsymbol{X}^\top\boldsymbol{X})b_0 + \frac{\eta_1}{n}\boldsymbol{X}^\top Y_\perp\right] =: r\cdot c_1 + d_1.$$

Next, at each time $t=2,\ldots,T$, assume that $\widetilde{\beta}_{t-1} = c_{t-1}r + d_{t-1}$. Then, writing $\mathcal{P}_{\mathcal{S}_{t-1}}$ as the matrix in $\mathbb{R}^{p\times p}$ which acts as the identity on groups in $\mathcal{S}_{t-1}$ and sets all other groups to zero, we have $b_{t-1} = \mathcal{P}_{\mathcal{S}_{t-1}}\widetilde{\beta}_{t-1} = \mathcal{P}_{\mathcal{S}_{t-1}}c_{t-1}r + \mathcal{P}_{\mathcal{S}_{t-1}}d_{t-1}$, and so

$$\widetilde{\beta}_t = b_{t-1} - \eta_t\nabla f(b_{t-1}) = b_{t-1} - \eta_t\left(\frac{1}{n}\boldsymbol{X}^\top(\boldsymbol{X}b_{t-1} - z_Y(r))\right)$$

$$= r\cdot\left[(\boldsymbol{I}_p - \frac{\eta_t}{n}\boldsymbol{X}^\top\boldsymbol{X})\mathcal{P}_{\mathcal{S}_{t-1}}c_{t-1} + \frac{\eta_t}{n}\boldsymbol{X}^\top U\right] + \left[(\boldsymbol{I}_p - \frac{\eta_t}{n}\boldsymbol{X}^\top\boldsymbol{X})\mathcal{P}_{\mathcal{S}_{t-1}}d_{t-1} + \frac{\eta_t}{n}\boldsymbol{X}^\top Y_\perp\right] =: r\cdot c_t + d_t.$$

### A.4 Simulation results for group lasso and forward stepwise regression

The group lasso is run with penalty parameter $\lambda = 4$. The group lasso algorithm is run via the R package `gglasso` [1]. Figure 1 shows the p-values obtained with the group lasso, while Figure 2 displays the coverage for the norms $\|\mathcal{P}_\mathcal{L}\mu\|_2$ and the inner products $\langle\text{dir}_\mathcal{L}(Y),\mu\rangle$; these plots are produced exactly as Figures 1 and 2 for IHT, except that only 200 trials are shown for the coverage plot due to the slower run time of this method. We observe very similar trends for this method as for IHT.

The forward stepwise method is implemented with $T=10$ many steps, and p-values and confidence intervals are computed by considering all 10 selected groups simultaneously at the end of the procedure (rather than sequentially) so that the results are more comparable to the other two methods. Figure 3 shows the p-values obtained with the forward stepwise method, while Figure 4 displays the coverage for the norms $\|\mathcal{P}_\mathcal{L}\mu\|_2$ and the inner products $\langle\text{dir}_\mathcal{L}(Y),\mu\rangle$; these plots are produced exactly as Figures 1 and 2 for IHT. We again observe similar trends in the results.

Figure 1: Group lasso. For each group, we plot its p-value for each trial in which that group was selected, for 200 trials. Histograms of the p-values for true signals (left, red) and for nulls (right, gray) are attached.

Figure 2: Group lasso. Empirical coverage over 200 trials with signal strength $\tau$. "Norm" and "inner product" refer to coverage of $\|\mathcal{P}_{\mathcal{L}}\mu\|_2$ and $\langle \text{dir}_{\mathcal{L}}(Y), \mu \rangle$, respectively.

## A.5  Pseudo-code: post-selection inference for forward selection

In Algorithm 1 we provide a detailed pseudo-code of our inference method for forward selection (described in Section 3.2). Here, we compute the $P$ value as well as confidence interval for each selected group conditioned on our previous selections. The algorithm is efficient and only overhead above the Forward Selection method is computation of the integral of a one-dimensional density over different intervals (see Step 15).

## References

[1] Yi Yang and Hui Zou. *gglasso: Group Lasso Penalized Learning Using A Unified BMD Algorithm*, 2014. URL `https://CRAN.R-project.org/package=gglasso`. R package version 1.3.

Figure 3: Forward stepwise regression. For each group, we plot its p-value for each trial in which that group was selected, for 200 trials. Histograms of the p-values for true signals (left, red) and for nulls (right, gray) are attached.

Figure 4: Forward stepwise regression. Empirical coverage over 2000 trials with signal strength $\tau$. "Norm" and "inner product" refer to coverage of $\|\mathcal{P}_{\mathcal{L}}\mu\|_2$ and $\langle \mathrm{dir}_{\mathcal{L}}(Y), \mu \rangle$, respectively.

---

**Algorithm 1** Post-selection Inference for Forward Selection

---

1: **Input :** Response $Y$, design matrix $\boldsymbol{X}$, groups $\mathcal{C}_1, \ldots, \mathcal{C}_G \subseteq \{1, \ldots, p\}$, maximum number of selected groups $T$, desired accuracy $\alpha$

2: **Initialize :** $\mathcal{S}_0 = \varnothing$, residual $\widehat{\epsilon}_0 = Y$, $\mathcal{R}_Y = \mathbb{R}_+$

3: **for** $t = 1, 2, \ldots, T$ **do**

4:     $g_t = \arg\max_{g \in [G] \setminus \mathcal{S}_{t-1}} \{\|\boldsymbol{X}_g^\top \widehat{\epsilon}_{t-1}\|_2\}$

5:     Update the model, $\mathcal{S}_t = \{g_1, \ldots, g_t\}$, and the residual, $\widehat{\epsilon}_t = \mathcal{P}_{\boldsymbol{X}_{\mathcal{S}_t}}^{\perp} Y$

6:     $\mathcal{L}_t \leftarrow \mathrm{span}(\mathcal{P}_{\boldsymbol{X}_{\mathcal{S}_{t-1}}}^{\perp} \boldsymbol{X}_{g_t})$, $U_t \leftarrow \dfrac{\mathcal{P}_{\mathcal{L}_t} Y}{\|\mathcal{P}_{\mathcal{L}_t} Y\|_2}$, $Y_\perp^t \leftarrow \mathcal{P}_{\mathcal{L}_t}^{\perp} Y$

7:     **for** $g \notin \mathcal{S}_t$ **do**

8:        $a_{t,g} \leftarrow \|\boldsymbol{X}_{g_t}^\top \mathcal{P}_{\boldsymbol{X}_{\mathcal{S}_{t-1}}}^{\perp} U_t\|_2^2 - \|\boldsymbol{X}_g^\top \mathcal{P}_{\boldsymbol{X}_{\mathcal{S}_{t-1}}}^{\perp} U_t\|_2^2$

9:        $b_{t,g} \leftarrow \langle \boldsymbol{X}_{g_t}^\top \mathcal{P}_{\boldsymbol{X}_{\mathcal{S}_{t-1}}}^{\perp} U_t, \boldsymbol{X}_{g_t}^\top \mathcal{P}_{\boldsymbol{X}_{\mathcal{S}_{t-1}}}^{\perp} Y_\perp^t \rangle - \langle \boldsymbol{X}_g^\top \mathcal{P}_{\boldsymbol{X}_{\mathcal{S}_{t-1}}}^{\perp} U_t, \boldsymbol{X}_g^\top \mathcal{P}_{\boldsymbol{X}_{\mathcal{S}_{t-1}}}^{\perp} Y_\perp^t \rangle$

10:       $c_{t,g} \leftarrow \|\boldsymbol{X}_{g_t}^\top \mathcal{P}_{\boldsymbol{X}_{\mathcal{S}_{t-1}}}^{\perp} Y_\perp^t\|_2^2 - \|\boldsymbol{X}_g^\top \mathcal{P}_{\boldsymbol{X}_{\mathcal{S}_{t-1}}}^{\perp} Y_\perp^t\|_2^2$

11:       $\mathcal{I}_{t,g} \leftarrow \{r \in \mathbb{R}_+ : a_{t,g} r^2 + 2 b_{t,g} r + c_{t,g} \geq 0\}$

12:       $\mathcal{R}_Y \leftarrow \mathcal{R}_Y \cap \mathcal{I}_{t,g}$

13:     **end for**

14:     $P_t = \dfrac{\int_{r \in \mathcal{R}_Y, r > \|\mathcal{P}_{\mathcal{L}_t} Y\|_2} r^{k-1} e^{-r^2/2\sigma^2} \, \mathrm{d}r}{\int_{r \in \mathcal{R}_Y} r^{k-1} e^{-r^2/2\sigma^2} \, \mathrm{d}r}$

15:     $L_\alpha^t = \beta$ s.t. $\dfrac{\int_{r \in \mathcal{R}_Y, r > \|\mathcal{P}_{\mathcal{L}_t} Y\|_2} r^{k-1} e^{-(r^2 - 2r\beta)/2\sigma^2} \, \mathrm{d}r}{\int_{r \in \mathcal{R}_Y} r^{k-1} e^{-(r^2 - 2r\beta)/2\sigma^2} \, \mathrm{d}r} = \alpha$

16: **end for**

17: **Output :** Selected groups $\{g_1, \ldots, g_T\}$, p-values $\{P_1, \ldots, P_T\}$, confidence interval lower bounds $\{L_\alpha^1, \ldots, L_\alpha^T\}$

---