[Reviews · NeurIPS 2016]

Reviewer 1

Summary

The paper proposes an approach to construct confidence intervals and p-values for the selection of groups of covariates. The approach is applicable to a broad class of methods, such as forward selection, iterative hard-thresholding, and group lasso, which are taken as illustrative examples in the paper. The challenge of the addressed problem is related to the fact that the statistical assessment of the importance of the groups must account for the fact that the same data have been used to select the groups. The novelty of the paper lies in the ability to construct confidence intervals while previous approaches for groups of covariates were limited to p-values. The contribution relies on a key lemma describing the distribution of truncated projections of Gaussian's. Two small-scale experiments are carried out, on synthetic and real-world data.

Qualitative Assessment

Clarity/Presentation/Related work/Presentation of contribution(s): The paper is overall clearly and well written. Also, the paper is well structured and makes a good introduction to the problem at hand, exposing what the new challenges and the resulting contributions are (e.g., paragraph lines 103-115). At some few places (see details below), some additional details would be useful. Technical level: The paper appears as technically sound and Lemma 1/Theorem 1 represent a non-trivial contribution. Some questions related to the proofs (see afterwards) should be clarified. Experiments: The experimental section may appear as a bit disappointing. For instance, it is unclear what the message of the second experiment (Section 4.2) is? Moreover, some further comparisons/discussions to other approaches may be beneficial (see details below). For example, since the main challenge of this work is to try to work with the same data as those used for the selection of the groups, it would make sense to compare against the more naive strategy based on splitting the data (which is supposed to lead to some loss of accuracy for model selection and power for inference). Details comments: * It should be discussed earlier in the paper whether the groups are assumed to form a partition (i.e., no overlaps). * A discussion about going beyond the quadratic loss (or equivalently, the Gaussian model) would be interesting. * Condition (1) should be more detailed: Why is this the right quantity to consider? * In Theorem 1, it would useful to have more details regarding the quantities we condition on and with respect to which random variables the probabilities are considered. * How is the equation hat{f_Y}(L_alpha) = alpha numerically solved? (line 252) * A discussion about the computational complexity of the approach in the different scenarios (IHT, group lasso, etc.) would be useful. * It is unclear to understand what the take-home message of the second experiment (Section 4.2) is? * Could the proposed methodology be used for techniques like [a, and references therein]? Moreover, how would the proposed approach compare with a simple bootstrap operation (along the lines of [b] but for group-sparse estimators)? * Supplementary material: In the beginning of the proof of Theorem 1, don't we need as well the continuity of t -> f_y(t) to guarantee the existence of L_alpha? More details should also be provided regarding the proofs of the properties of f_y (e.g., limiting behaviors and monotonicity). [a] Ndiaye, E.; Fercoq, O.; Gramfort, A. & Salmon, J. GAP Safe screening rules for sparse multi-task and multi-class models Advances in Neural Information Processing Systems, 2015, 811-819 [b] Bach, F. Bolasso: model consistent Lasso estimation through the bootstrap Proceedings of the International Conference on Machine Learning (ICML), 2008

Confidence in this Review

2-Confident (read it all; understood it all reasonably well)


Reviewer 2

Summary

The authors propose a framework for evaluating the significance and the associated confidence interval for a selected group of variables in the context of group-sparse regression.

Qualitative Assessment

I found this paper quite difficult to read. I'm afraid this as well as the high density of the paper has impaired my understanding of the contribution. In particular, I am not very clear on the incremental contribution over Loftus and Taylor, 2015. As far as I understand, the goal of the paper is to provide a way to test, either iteratively or as a whole, the relevance of groups of variables selected by group-sparse linear algorithms. The authors mainly focus on three such algorithms. While I can clearly see the advantages of being able to compute these significance results, I am afraid that I can't provide a relevant review for this paper and for this I apologize, hoping that my fellow reviewers will feel more confident. Let me still try to focus on the applications. In the California data set example and table 1, I don't understand how the 3-variable groups are constructed. I was actually hoping to see actual groups of variables but I understand that the 3-variable groups are constructed by "expanding" one variable into 3? Why not let the algorithms actually choose groups of variables? I'm assuming this might be for comparison purposes but it also makes the results somewhat unclear. In general, I would have liked to see a more algorithmic version of the procedure somewhere in the paper. From what I see here, I am really not sure how this method is/could be implemented.

Confidence in this Review

1-Less confident (might not have understood significant parts)


Reviewer 3

Summary

The authors consider valid inference for the effect size in linear regression associated with groups of variables selected after viewing the data. A selective test for the null hypothesis that the projection of the mean onto the linear space spanned by the group variables, has been suggested before, but the existing test is not useful for obtaining a confidence interval. By deriving the exact distribution of this test statistic in the "noncentral" case (i.e., without assuming the null is true), a selective confidence interval is obtained for the size of the projection. Inference is in fact provided for the size of the projection of \mu onto the vector which is the projection of Y onto the linear space (this is indeed a parameter once conditioning on that direction).

Qualitative Assessment

I found the paper to be an important extension of the previous paper by Loftus and Taylor ('15). The difference between the null distribution of the considered test statistic and the non-null distribution is quite interesting, and the technical problems that arise are substantial. A few comments: 1. Is there anything that can be said about properties of the proposed CI? in the case of inference for the coefficient of a selected variable in Gaussian linear regression (no groups), the conditioning on nuisance statistics is justified by theory for general exponential families (Fithian et al, '15).. 2. I understand that conditionally on dir_L(Y), the distribution of the test statistic is governed by < dir_L(Y), \mu > rather than < dir_L(\mu), \mu >=\| P_L \mu \|. The proposed CI is guaranteed to cover \| P_L \mu \| w.p. 1-alpha conditionally on dir_L(Y) but is usually conservative; but it then follows that also *unconditionally* on dir_L(Y), the CI should be valid for the parameter of interest. In that case maybe the CI is less conservative? 3. Consider modifying the wording in the following: -line 59: "distribution of subsets 60 of parameters " (please be more precise; this sentence doesn't make sense) -line 132: "Lemma 1 yields a truncated Gaussian distribution"

Confidence in this Review

2-Confident (read it all; understood it all reasonably well)


Reviewer 4

Summary

This paper studies the (relatively recently introduced) problem of selective inference for linear models with group sparsity. The authors provide new estimators for confidence intervals and testing selected groups.

Qualitative Assessment

The paper is well-written and seems like an interesting contribution to this area (though I am not familiar with previous work beyond what the authors cite). The theoretical results are non-trivial and the experimental evaluation sounds promising.

Confidence in this Review

1-Less confident (might not have understood significant parts)


Reviewer 5

Summary

This paper presents a novel method for selective inference in the setting of group sparsity that includes the construction of confidence intervals and p-values for testing selected groups of variables. The originality lies in the development of the projection of the data onto a given space and inference procedures for a broad class of group-sparse selection methods including the group lasso, iterative hard thresholding, and forward stepwise regression.

Qualitative Assessment

The proposed method is practically useful and can be accepted to the conference.

Confidence in this Review

2-Confident (read it all; understood it all reasonably well)


Reviewer 6

Summary

This paper proposes an approach to perform group-sparse inferences for the (sparse) feature selection methods using the conditional probability density \|P_\mathcal{L}Y\|_2, i.e. to provide p-values and confidence intervals for the selected features/coefficients. Three sparse regression methods are then studied and algorithms for performing the inferences on all three methods are given respectively, followed by experiments on simulated and real data.

Qualitative Assessment

1. The proposed approach performs inferences with respect to groups of columns of X. Is there any advantage/motivation to perform inferences in groups instead of doing it one by one column? 2. In practice, how do you decide which columns should be grouped together? 3. Are the groups mutually exclusive? 4. Line 61-63: Can you provide details on how to "control for the remaining selected groups"? 5. Line 109-110: What do you mean by saying "condition on the direction dir_\mathcal{L}(Y)"? 6. Line 113-114: What do you mean by saying "independence between the length and the direction of P_\mathcal{L}Y"? When P_\mathcal{L}\mu\neq0, why do the two become dependent? 7. Line 255: The matrix X is 500 by 500, right? Do the columns in each group share some similar structure? If the entries of X are i.i.d. generated, how do you decide which columns should be grouped together? 8. Line 279-280: For the real data, how do you choose the group and the group size? Why do you expand one column to a group of three instead of combining different columns? This doesn't produce anything new. 9. How do you evaluate the performance? Is there a baseline or some other inference method that you can compare the proposed approach against? Can you compare the proposed approach with the method from reference 6? 10. For the real data, since each group contains only information from one column of the original data, how come that the p-values differ so much using the same method? Take the "Forward stepwise" for example, when group size is 3, the p-value of "Obese" is 0, which is quite different from 0.519 when group size is 1. Which one is more accurate?

Confidence in this Review

2-Confident (read it all; understood it all reasonably well)